# ACADEMICEVAL: LIVE LONG-CONTEXT LLM BENCHMARK

## ABSTRACT

Large Language Models (LLMs) have achieved remarkable performance in long-context understanding. However, current long-context LLM benchmarks are limited by rigid context length and labor-intensive annotation, and the label leakage issue in LLM training also poses a pressing challenge. Therefore, we propose ACADEMICEVAL, a live benchmark for evaluating LLMs over long-context generation tasks. ACADEMICEVAL adopts papers on arXiv to introduce several academic writing tasks with long-context inputs, *i.e.*, TITLE, ABSTRACT, INTRODUCTION, and RELATED WORK, which cover a wide range of abstraction levels and require no manual labeling. Moreover, ACADEMICEVAL integrates high-quality and expert-curated few-shot demonstrations from a collected co-author graph to enable flexible context length. Especially, ACADEMICEVAL features an efficient live evaluation, ensuring no label leakage. We conduct holistic experiments on ACADEMICEVAL, and the results illustrate that LLMs perform poorly on tasks with hierarchical abstraction levels and tend to struggle with long few-shot demonstrations, illustrating the challenge of our benchmark. We also provide insightful analysis for enhancing LLMs' long-context modeling capabilities.

## 1 INTRODUCTION

Large Language Models (LLMs) have recently achieved tremendous success in natural language processing (NLP) tasks Achiam et al. (2023); AI@Meta (2024). However, when facing long context inputs, LLMs show a sharp decline in performance, which poses a pressing challenge to LLMs in understanding and capturing key information in long texts Li et al. (2024); Liu et al. (2024). Therefore, several long-context LLM benchmarks are spawned to evaluate LLMs in various settings, including question answering, summarizing, and reasoning Shaham et al. (2023); An et al. (2023); Dong et al. (2023); Bai et al. (2023b); Li et al. (2023); Zhang et al. (2024). Despite their success, these benchmarks still suffer from concerns of rigid context length, saturated performance, and being leaked in LLM training.

We envision that the *next-generation long-context LLM benchmarks* should ideally possess three key features. (1) *Flexible* and potentially *unlimited* context length: existing benchmarks fix the context for each long-context problem; ideally, the format and length of the context could be flexibly set based on the LLM's capability, especially given the release of long-context LLMs Reid et al. (2024) and their capabilities in ingesting multi-modal information, *e.g.*, graphs Dong et al. (2024). (2) High-quality labels derived from *real-world data*, *minimizing* human labeling efforts: existing long-context benchmarks often require human labeling Bai et al. (2023b); An et al. (2023); Li et al. (2023); Dong et al. (2023); Zhang et al. (2024), which is costly and limits the size of the benchmarks to about 2000 samples Xu et al. (2023) (3) Live updates to mitigate information leakage during LLM pretraining and fine-tuning: benchmark data contamination in LLM has gradually become a severe issue Sainz et al. (2023); Ye et al. (2024); Zhu et al. (2024b;a); Xu et al. (2024); we argue that holding out future data as the val/test set is one of the most effective approaches for open benchmarks.

Based on these principles, we propose ACADEMICEVAL, a live benchmark to evaluate LLMs over long-context generation tasks. ACADEMICEVAL adopts arXiv as its data source and features a suite of academic writing tasks on each paper without labor-intensive annotation: TITLE, ABSTRACT, INTRODUCTION, and RELATED WORK, each of which has long-context input and hierarchical abstraction levels. In particular, we construct a co-author graph via arXiv API to conveniently obtain

Table 1: **Comparison with Existing Long-context LLM Benchmarks**. Each column indicates the average input length, whether the annotation is human-assisted, whether there are tasks with hierarchical abstraction levels, whether it contains few-shot demonstrations, and whether the benchmark is lively updated, respectively.

| Benchmark | Avg Len | Automatic Annotation | Hierarchical Abstraction | Few-shot Demons | Live Update |
|---|---|---|---|---|---|
| ZeroSCROLLS (Shaham et al., 2023) | ∼10K | ✓ | ✗ | ✗ | ✗ |
| L-Eval (An et al., 2023) | ∼8K | ✗ | ✗ | ✗ | ✗ |
| BAMBOO (Dong et al., 2023) | ∼16K | ✗ | ✗ | ✗ | ✗ |
| LongBench (Bai et al., 2023b) | ∼8K | ✗ | ✗ | ✓ | ✗ |
| LooGLE (Li et al., 2023) | ∼20K | ✗ | ✗ | ✗ | ✗ |
| ∞Bench (Zhang et al., 2024) | ∼200K | ✗ | ✗ | ✗ | ✗ |
| **AcademicEval (ours)** | **Flexible** | ✓ | ✓ | ✓ | ✓ |

co-author papers as high-quality and expert-curated few-shot demonstrations, which also possess ACADEMICEVAL flexible context length. Furthermore, ACADEMICEVAL introduces efficient live evaluation based on the co-author graph, which utilizes the latest papers on arXiv to update the benchmark data periodically and ensures no label leakage. Moreover, ACADEMICEVAL provides in-context few-shot demonstrations for each data sample, which is neglected by most existing long-context LLM benchmarks Liu et al. (2024); Li et al. (2024). In the experiment, we evaluate three types of LLMs on ACADEMICEVAL: standard LLMs, long-context LLMs, and retrieval-augmented language models (RALM). Experimental results show that current LLMs cannot deal with long-context context tasks well at diverse abstraction levels, and RALM is the worst-performing one among the three types of baselines. Additionally, as the input length increases, noticeable performance degradation can be seen on almost all tasks, with the largest drop reaching 32% and 7% w.r.t. RougeL Lin (2004) and BERTScore Zhang et al. (2019), respectively. Although we find that few-shot demonstrations from co-author papers can slightly strengthen the performance over some tasks, it is still limited by the long context modeling capabilities of LLMs. In general, the experimental findings indicate that ACADEMICEVAL is a challenging long-context LLM benchmark.

We illustrate the comparison with existing long-context LLM benchmarks in Table 1. Our contributions are summarized as follows:

- We propose a live benchmark, ACADEMICEVAL, to evaluate LLMs over long-context generation tasks. ACADEMICEVAL features four academic writing tasks with hierarchical abstraction levels and requires no manual annotation.
- We construct a co-author graph via the arXiv API and draw on the co-author papers as informative few-shot demonstrations, making the context length of ACADEMICEVAL flexible and scalable.
- ACADEMICEVAL conducts periodic data updates on the co-author graph to enable efficient live evaluation, which ensures no label leakage and fair evaluation.
- We conduct comprehensive experiments on ACADEMICEVAL, demonstrating its challenges and providing insights for improving LLMs in long-context modeling.

## 2 RELATED WORK

**Long-context Modeling and LLM Benchmarks** LLMs are known to be powerful in language modeling tasks Achiam et al. (2023); AI@Meta (2024). However, when it comes to long-context input, LLMs show a sharp decline in performance, posing a pressing challenge when benchmarking their long-context modeling capabilities Liu et al. (2024); Li et al. (2024). Currently, there are two mainstream technologies for long-context modeling tasks: retrieval-augmented language models (RALM)Ram et al. (2023); Yu et al. (2023); Trivedi et al. (2022); Jiang et al. (2023); Asai et al. (2023) and long-context LLMs Bai et al. (2023a); Jiang et al. (2024); Teknium et al.. RALM equips LLMs with a retrieverRobertson et al. (2009); Ramos et al. (2003); Karpukhin et al. (2020); Izacard et al. (2021) to perform information retrieval on short text chunks, which are then fed to LLMs

together with the input query to generate the final output. As a retrieval system, RALM is usually evaluated over retrieval-based benchmarks, including STARK Wu et al. (2024), RGB Chen et al. (2024), ARES Saad-Falcon et al. (2023), etc. In comparison, long-context LLMs expand their context window length to accommodate longer inputs and are benchmarked over various tasks, which include long-context QA, summarization, conversations, reasoning, etc Shaham et al. (2023); An et al. (2023); Dong et al. (2023); Bai et al. (2023b); Li et al. (2023); Zhang et al. (2024).

**Label Leakage in LLM Benchmarks** Label leakage has always been a severe issue that benchmarks must attempt to avoid during data collection. However, recent researches Xu et al. (2024); Zhu et al. (2024b;a); Ye et al. (2024) point out that most LLM benchmarks are composed of statically collected data, which may be inevitably included in the large amount of training data of LLMs, causing label leakage. Therefore, some works attempt to measure or detect the extent of label leakage in LLM benchmarks. Benbench Xu et al. (2024) leverages perplexity and N-gram accuracy to quantify potential label leakage, while PAC Ye et al. (2024) detects contaminated data by comparing the polarized distance of samples before and after augmentation. Even though these approaches propose to measure or detect label leakage, there is little work on mitigating and solving this issue Zhu et al. (2024b). Dynabench Kiela et al. (2021) and Dynaboard Ma et al. (2021) feature dynamic human-in-the-loop dataset creation while avoiding leakage, which is very labor-intensive. DyVal Zhu et al. (2024b) leverages pre-set constraints and directed acyclic graphs (DAG) to dynamically generate test cases with diverse complexities, reducing the risk of label leakage. FreshBench Zhu et al. (2024a) and StackMIA Ye et al. (2024) collect the latest data from public websites periodically and simply rely on the chronological split to build a dynamic benchmark.

**Long-context Summarization Benchmarks** Solving ACADEMICEVAL requires LLM's long-context summarization capability Liu et al. (2024). Existing works include (1) query-based summarization tasks, focusing on the capability of models to position and capture local key information in long texts given a specific query Litvak & Vanetik (2017); Wang et al. (2022); (2) single-document or multi-document summarization tasks concentrate on evaluating the ability of models to understand long texts holistically Cohan et al. (2018); Meng et al. (2021); Huang et al. (2021); Kryściński et al. (2021); Cachola et al. (2020). These long-context summarization benchmarks suffer from the above-mentioned limitations, including requiring human-assisted labeling and concerns about data leakage; moreover, these summarization tasks focus on one-level summarization, failing to consider the summarizations at different abstraction levels.

## 3 ACADEMICEVAL BENCHMARK

In this section, we propose ACADEMICEVAL (Figure 1) for live evaluation in long-context generation tasks with hierarchical abstraction levels. We first describe data collection and preprocessing in Section 3.1. Then, in Section 3.2, four academic writing tasks with diverse abstraction levels are introduced, and we also integrate few-shot demonstrations to make the context length flexible and scalable. Finally, Section 3.3 elucidates the live evaluation with periodic data updates.

### 3.1 DATA CURATION

**Co-author Graph Construction via arXiv** As a public paper preprint platform, arXiv[1] has always been favored by researchers. It archives a huge amount of papers and updates the latest ones daily, which serves as an excellent data source and also lays the foundation for the live update of our benchmark. Thanks to the arXiv API[2], paper files can be obtained in batch without much manual effort. We first collect and construct a co-author graph (edges are established between two author nodes that are co-authors in at least one paper) using the arXiv API through breadth-first search (BFS), where the features of each author node include the first-author papers published by the author. By making the co-author graph the carrier of paper data, we can form an interconnected whole of scattered papers, which provides valuable structural information to be exploited for our benchmark (*e.g.*, as few-shot demonstrations). Furthermore, we can enable efficient live updates on the co-author graph, which will be introduced in Section 3.3.

---

[1]https://arxiv.org/

[2]https://info.arxiv.org/help/api/index.html

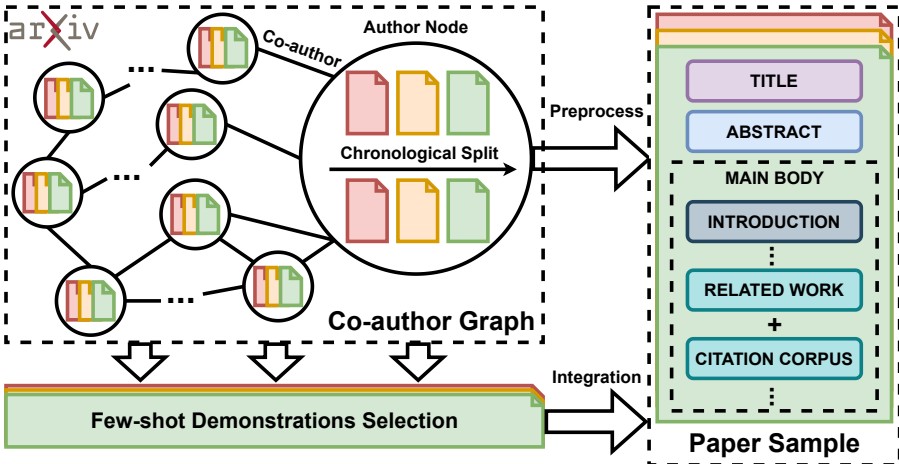

Figure 1: **AcademicEval Benchmark.** We construct a co-author graph via arXiv and conduct a chronological split on all paper samples (training, validation, and test samples are represented by red, orange, and green, respectively). Each paper sample is preprocessed into separate sections and can be integrated with few-shot demonstrations from co-author papers.

**Academic Data Gathering and Preprocessing** After some basic operations on the co-author graph, such as taking the maximum connected component, we preprocess all papers in the node features. For each paper, we collect related metadata via the arXiv API, including author information, publication timestamp, etc., and download the pdf file simultaneously. Intuitively, the paper content can be considered as a kind of original, expert-curated, and high-quality labeled data without manual annotation. Therefore, we develop a complete pipeline for preprocessing each paper, splitting and extracting the text of several sections in it. To give an example, we can utilize the pipeline to extract the introduction section from the main body of a paper. Then, the extracted introduction section, the remaining parts of the extracted main body (*i.e.*, w/o the introduction section), and the abstract and title constitute the basic data of each paper sample. For the related work section, we extract each cited paper's abstract and title to form an additional citation corpus.

We will describe in detail in Section 3.2 how to use these data to design long-context academic writing tasks.

### 3.2 BENCHMARKING LLMs OVER LONG-CONTEXT GENERATION TASKS WITH HIERARCHICAL ABSTRACTION

**Task Description** Employing machine learning approaches to automate academic writing has always been a research hotspot with significant practical application value Chen et al. (2022; 2021). Therefore, inspired by the leave-one-out validation, we introduce four academic writing tasks with ultra-long context to evaluate the generation capability of LLMs under different abstraction levels, as shown below:

- TITLE WRITING. This task takes a paper's main body and abstract, along with a specific task prompt as inputs, and then asks LLMs to output a predicted title.

- ABSTRACT WRITING. Similar to the above, this task takes a paper's main body and title, along with a specific task prompt as inputs, and then asks LLMs to output a predicted abstract.

- INTRODUCTION WRITING. This task takes a paper's main body (with the introduction section removed), title, and abstract, along with a specific task prompt as inputs, and then asks LLMs to output a predicted introduction.

- RELATED WORK WRITING. This task takes a paper's main body (with the related work section removed), title, abstract, and citation corpus (introduced in Section 3.1), along with a specific task prompt as inputs, and then asks LLMs to output a predicted related work.

Based on the above task descriptions, we can generate four basic benchmark settings with different abstraction levels, namely TITLE-10K, ABS-9K, INTRO-8K and RELATED-34K, with suffixes indicating their input context length[3].

**Integration of Few-shot Demonstrations** Given the rigid context length of current long-context LLM benchmarks and the general effectiveness of in-context learning in LLMs Dong et al. (2022); Wei et al. (2022a;b); Kojima et al. (2022), we propose to integrate long few-shot demonstrations to enable flexible and scalable context length, and we have two selection options for each sample in the above four basic benchmark settings: *(1) Randomly select papers under the same category*. According to the paper categories provided by the arXiv API, we can randomly select several non-duplicate papers under the same category. *(2) Randomly Select co-author papers*. The motivation is straightforward: the similarity of research directions between co-author papers is more fine-grained. Thanks to the co-author graph, it is convenient to obtain co-author papers of each original paper sample. These selected papers serve as few-shot demonstrations and are utilized as input-output pairs to enrich the input context of the original samples, providing potentially insightful and relevant content while enabling flexible and scalable context length.

Consequently, we have completed the construction of benchmark settings, and the data statistics in the initial collection round are shown in Table 2.

**Data Statistics** As shown in Table 2, ACADEMICEVAL has four academic writing tasks with hierarchical abstraction levels, and each task features four settings with diverse input context lengths, some of which are obtained by integrating few-shot demonstrations. For instance, each sample in TITLE-10K consists of a single paper sample. TITLE-30K and TITLE-31K-G are obtained by integrating with two few-shot demonstrations from random papers and co-author papers, respectively, while TITLE-50K-M is obtained by using both of the above integration options. Actually, we can scale context length by increasing the number of few-shot demonstrations to provide more informative references, enhancing task performance.

Furthermore, we present the text compression rate (defined as the number of input tokens divided by the number of output tokens) for each benchmark setting in Table 2 to illustrate the diverse abstraction levels in ACADEMICEVAL. Across the four tasks, a higher compression rate means a higher level of text abstraction in this task. Among several settings within each task, a higher compression rate makes it tougher to exploit information holistically but more likely to produce better outputs (since more references are integrated). These different tasks and settings increase the diversity of the ACADEMICEVAL benchmark.

As for data splitting, we perform a chronological split in ACADEMICEVAL, which means that the test set always contains the latest papers collected in each collection round, ensuring no label leakage. Note that Table 2 shows only the data collected in the initial round, which will be updated periodically as described in the next section.

### 3.3 LIVE EVALUATION WITH PERIODIC DATA UPDATES ON THE CO-AUTHOR GRAPH

The daily updates of arXiv provide the basis for the live evaluation of ACADEMICEVAL: we can periodically update the benchmark with the latest papers on arXiv. By setting a reasonable update cycle (*e.g.*, monthly or quarterly), we can ensure that the data in the benchmark is not contaminated so that it can be used to evaluate LLMs fairly in a live manner. Therefore, we proposed an efficient incremental update procedure on the co-author graph:

**(1) Node Update** For each author on the co-author graph, check whether the author has a newly published first-author paper through the arXiv API. If so, add it to the corresponding node feature on the co-author graph.

**(2) Node and Edge Update** During the traversal of Node Update, each author's new co-authors are added to a candidate list, and the number of new papers (including first-author and non-first-author papers) when searching for the author is used as the priority of the co-authors (co-authors of active authors tend to be active as well, and we can efficiently collect the latest papers from active authors).

---

[3]We use BERT Devlin et al. (2018) tokenizer by default to count the number of input tokens (output tokens are not included).

Table 2: **Data Statistics of AcademicEval (Initial Round).** It includes 4 writing tasks and provides four settings of different context length for each task. For each setting, we list their Comp. Rate, Samples of Each, Chronological Split, and Timespan of Test Data.

| Setting | Comp. Rate (In-Len. / Out-Len.) | #Samples of Each. | Chronological Split (Train-Val-Test) | Timespan of Test Data |
|---|---|---|---|---|
| **TITLE WRITING** | | | | |
| **TITLE-10K** | 587 | | | |
| **TITLE-30K** | 1773 | 5098 | 72%-19%-9% | 2024.06- |
| **TITLE-31K-G** | 1807 | | | 2024.07 |
| **TITLE-50K-M** | 2968 | | | |
| **ABSTRACT WRITING** | | | | |
| **ABS-9K** | 36 | | | |
| **ABS-28K** | 108 | 5098 | 72%-19%-9% | 2024.06- |
| **ABS-29K-G** | 112 | | | 2024.07 |
| **ABS-48K-M** | 185 | | | |
| **INTRODUCTION WRITING** | | | | |
| **INTRO-8K** | 6 | | | |
| **INTRO-28K** | 21 | 4665 | 71%-20%-9% | 2024.06- |
| **INTRO-28K-G** | 22 | | | 2024.07 |
| **INTRO-48K-M** | 37 | | | |
| **RELATED WORK WRITING** | | | | |
| **RELATED-34K** | 34 | | | |
| **RELATED-53K** | 53 | 2240 | 72%-20%-8% | 2024.06- |
| **RELATED-53K-G** | 53 | | | 2024.07 |
| **RELATED-72K-M** | 72 | | | |

Note: We use the BERT tokenizer by default to count the number of tokens.

Then, we use the prioritized candidate list to conduct BFS to update nodes and edges until a specific number of incremental update papers is met.

**(3) Graph Pruning** As the benchmark is updated, we will remove some outdated papers and inactive authors (defined as those who have not published new first-author or non-first-author papers for a long time) from the co-author graph.

In this way, the latest papers can be obtained sufficiently and efficiently while ensuring connectivity and a smaller graph size.

**Live Leaderboard** We also provide a leaderboard for live evaluation of the current most advanced LLMs, which will be released later.

## 4 EXPERIMENTS

### 4.1 BASELINES

We adopt the following three types of baselines to conduct a holistic evaluation of ACADEMICEVAL.

**Standard LLMs** We choose Gemma Instruct (7B) Team et al. (2024) and LLaMA-3 Chat (70B) AI@Meta (2024) as standard LLM baselines, each with a context length of 8K.

**Long-context LLMs** We choose Qwen 1.5 Chat (72B) Bai et al. (2023a), Mixtral-8x7B Instruct (46.7B) Jiang et al. (2024), and Nous Hermes 2 - Mixtral 8x7B-DPO (46.7B) Teknium et al. as long-context LLM baselines, each with a context length of 32K.

**Retrieval-augmented language models (RALM)** First, we consider two sparse retrievers: (1) **BM25** Robertson et al. (2009): This is a widely used retrieval model that ranks documents based on the frequency of query terms in each document. (2) **TF-IDF** Ramos et al. (2003): It scores documents by multiplying the term frequency of each query term by the inverse document frequency. Second, we also consider three dense retrievers: (3) **DPR** Karpukhin et al. (2020): It uses a bi-encoder to retrieve relevant documents based on dense embeddings. (4) **Contriever** Izacard et al. (2021): It leverages unsupervised contrastive learning to learn high-quality dense representations. (5) **Dragon** Lin et al. (2023): It enhances retriever training by employing data augmentation, including query and label augmentation.

We use the inputs of ACADEMICEVAL as the external corpus of RALM. For text split, we use the RecursiveCharacterTextSplitter from LangChain[4] and set chunk size and chunk overlap to 512 and 64, respectively. For each retrieval, we recall up to 12 text chunks (limited by the context length of standard LLMs) based on text similarity (semantic similarity based on inner product for dense retrievers or similarity based on word frequency for sparse retrievers).

## 4.2 EVALUATION METRICS

For evaluation metrics, we adopt (1) **BERTScore**[5] Zhang et al. (2019): This metric leverages BERT-based embedding to measure semantic similarity between predicted and reference texts. (2) **RougeL** Lin (2004): This metric evaluates the longest common subsequence between the generated and reference texts, providing a measure of similarity in terms of sequential matching. For both metrics, higher scores indicate a better match between the predicted and the reference text.

## 4.3 MAIN RESULT ANALYSIS

We conduct comprehensive experiments on the four academic writing tasks, and the results w.r.t. BERTScore and RougeL are presented in Table 3 and 4, respectively. Note that we do not conduct experiments on -M settings because its context length is too long for most of our selected baselines.

**Diverse Task Difficulties and Abstractions** The four tasks we proposed are designed to challenge LLMs over long-context generation tasks with different abstraction levels. From Table 3 and 4, we can clearly observe that it provides different difficulties for LLMs to perform well from TITLE WRITING to RELATED WORK WRITING tasks, and the results of all baselines on these four tasks have a relatively obvious trend. For example, the TITLE WRITING task tends to have a higher score than the ABSTRACT WRITING task, which may indicate that the TITLE WRITING task is easier than the ABSTRACT WRITING task. Since a title only has a few words, LLMs only need to generate a roughly related theme to achieve a high semantic similarity, while an abstract requires a more detailed description generated to achieve it.

**Baseline Performance Comparison** Among different baselines, RALM with LLaMA frequently delivers the highest scores across various tasks and context lengths, with only a context length of 8K. Standard LLMs also achieve competitive performance, which is slightly inferior to long-context LLMs. This exposes the shortcomings of long-context LLMs' generation capabilities, which is well revealed by ACADEMICEVAL. Among long-context LLMs, Hermes performs best overall, but is still slightly inferior to RALM with LLaMA. This shows that although the current long-context LLMs have a longer context window size, they still have great deficiencies in processing long text information. In contrast, RALM-based methods generally outperform other baselines. This is primarily due to the retrieval mechanisms of RALM, which retrieves and processes information in a few relevant shorter chunks, enabling it to focus on key information.

**Impact of Context Length** The impact of context length on performance is evident across all task settings and both metrics, with baselines generally performing worse as the context length increases. For example, the TITLE WRITING task shows a noticeable drop in scores as the context length extends from 10K to 31K tokens. This trend is also apparent in ABSTRACT WRITING and INTRODUCTION WRITING, where longer contexts correlate with decreased model performance. showing that our benchmark challenges LLMs in effectively processing ultra-long inputs.

---

[4]https://www.langchain.com/

[5]We use deberta-xlarge-mnli He et al. (2021) instead of the default roberta-large Liu et al. (2019) as the backbone model to have the best correlation with human evaluation.

Table 3: **Main Results on AcademicEval w.r.t. BERTScore.**

| Models | Standard LLMs | | Long-context LLMs | | | RALM | |
|---|---|---|---|---|---|---|---|
| | Gemma | LLaMA | Qwen | Mixtral | Hermes | Gemma$^\dagger$ | LLaMA$^\dagger$ |
| #Params. | 7B | 70B | 72B | 8x7B | 8x7B | 7B | 70B |
| Context Length | 8K | 8K | 32K | 32K | 32K | 8K | 8K |
| **Setting: TITLE WRITING** | | | | | | | |
| TITLE-10K | 66.1 | 74.1 | 73.9 | 73.4 | **74.2** | 65.8 | 73.9 |
| TITLE-30K | - | - | 73.0 | 72.9 | 73.4 | 65.7 | **73.9** |
| TITLE-31K-G | - | - | 72.8 | 72.8 | 73.3 | 65.7 | **73.8** |
| **Setting: ABSTRACT WRITING** | | | | | | | |
| ABS-9K | 59.9 | 62.4 | **62.5** | 61.4 | 62.2 | 60.3 | 61.5 |
| ABS-28K | - | - | 61.3 | 61.2 | **62.6** | 60.1 | 61.4 |
| ABS-29K-G | - | - | 61.3 | 61.4 | **62.5** | 60.2 | 61.3 |
| **Setting: INTRODUCTION WRITING** | | | | | | | |
| INTRO-8K | 54.8 | **55.8** | 55.4 | 54.6 | 55.2 | 55.0 | 55.2 |
| INTRO-28K | - | - | 54.8 | 54.0 | 54.8 | 55.0 | **55.2** |
| INTRO-28K-G | - | - | 54.9 | 54.1 | 54.7 | 55.0 | **55.3** |
| **Setting: RELATED WORK WRITING** | | | | | | | |
| RELATED-34K | 52.0 | 56.2 | **58.5** | 55.3 | 57.8 | 52.4 | 54.7 |
| RELATED-53K | - | - | - | - | - | 52.4 | **54.7** |
| RELATED-53K-G | - | - | - | - | - | 52.4 | **54.8** |

**Bold** indicates the highest score in each row.
† denotes augmentation with a retriever (Default: Contriever).
"-" means that the context length is too long to be fed into LLMs.

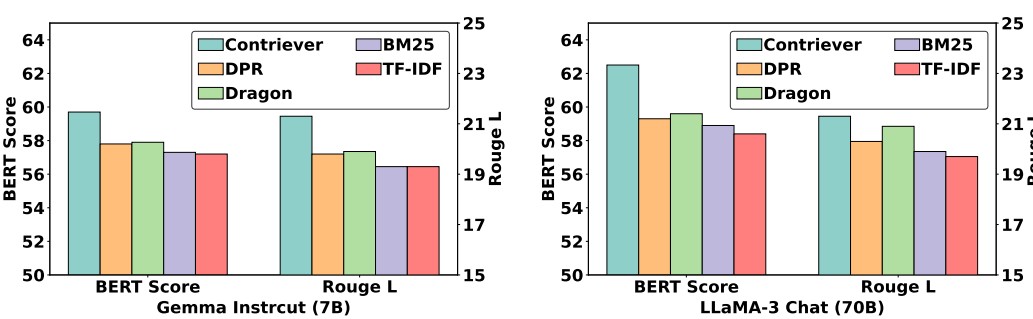

Figure 2: **Analysis of RALM on ABS-9K.** The left figure shows results with Gemma Instruct (7B), while the right one shows results with LLaMA-3 Chat (70B).

**Impact of Few-shot Demonstrations** From Table 3 and 4, we can observe that the integration of few-shot demonstrations generally degrades the performance of baselines, except for a few tasks where the results are slightly improved. This shows that current LLMs cannot exploit long few-shot demonstrations to benefit the target tasks well, emphasizing the importance of evaluating long in-context learning in LLM benchmarks. In addition, we can also find that few-shot demonstrations from co-author papers generally have a more positive impact on task performance than randomly selected ones.

Table 4: **Main Results on AcademicEval w.r.t. RougeL.**

| Models | Standard LLMs | | Long-context LLMs | | | RALM | |
|---|---|---|---|---|---|---|---|
| | Gemma | LLaMA | Qwen | Mixtral | Hermes | Gemma$^\dagger$ | LLaMA$^\dagger$ |
| #Params. | 7B | 70B | 72B | 8x7B | 8x7B | 7B | 70B |
| Context Length | 8K | 8K | 32K | 32K | 32K | 8K | 8K |
| **Setting: TITLE WRITING** | | | | | | | |
| TITLE-10K | 44.5 | 47.1 | 44.2 | 45.2 | 46.2 | 42.7 | **47.3** |
| TITLE-30K | - | - | 42.9 | 44.6 | 45.9 | 42.6 | **47.3** |
| TITLE-31K-G | - | - | 44.2 | 44.4 | 45.3 | 42.5 | **47.0** |
| **Setting: ABSTRACT WRITING** | | | | | | | |
| ABS-9K | 22.4 | 25.0 | 24.3 | 24.1 | **26.1** | 23.4 | 24.2 |
| ABS-28K | - | - | 23.3 | 24.7 | **26.6** | 23.1 | 24.1 |
| ABS-29K-G | - | - | 23.3 | 24.9 | **26.6** | 23.2 | 24.0 |
| **Setting: INTRODUCTION WRITING** | | | | | | | |
| INTRO-8K | 14.9 | **18.1** | 16.2 | 17.2 | 17.8 | 15.4 | 17.9 |
| INTRO-28K | - | - | 16.3 | 17.5 | 17.5 | 15.3 | **17.8** |
| INTRO-28K-G | - | - | 16.3 | 17.5 | 17.5 | 15.4 | **17.8** |
| **Setting: RELATED WORK WRITING** | | | | | | | |
| RELATED-34K | 13.5 | 14.9 | **16.0** | 13.4 | 15.1 | 14.1 | 15.3 |
| RELATED-53K | - | - | - | - | - | 14.0 | **15.3** |
| RELATED-53K-G | - | - | - | - | - | 14.0 | **15.2** |

**Bold** indicates the highest score in each row.

† denotes augmentation with a retriever (Default: Contriever).

"-" means that the context length is too long to be fed into LLMs.

## 4.4 ADDITIONAL ANALYSIS ON RALM

We conduct extensive experiments on RALM on the ABS-9K setting using standard LLMs Gemma Instruct (7B) and LLaMA-3 Chat (70B), and the results are presented in Figure 2. We can find that the performance of dense retrievers consistently outperforms sparse retrievers, among which contriever achieves the best results. This is because the summary generation task emphasizes semantic similarity, which can be well measured by the similarity of dense embeddings. However, the sparse retrievers perform text chunk recall based on sparse embeddings, and the results are significantly worse than those of the dense retrievers.

## 5 CONCLUSION

In this paper, we propose ACADEMICEVAL, a live long-context LLM benchmark for evaluating long-context generation tasks with hierarchical abstraction levels. ACADEMICEVAL adopts arXiv as the data source and introduces several long-context academic writing tasks without manual annotation since the papers on arXiv can be regarded as original, high-quality, and expert-curated labels. Moreover, we integrate few-shot demonstrations from a collected co-author graph to make the context length of our benchmark flexible and scalable. An efficient live evaluation is also designed to make ACADEMICEVAL immune to the label leakage issue and move toward a more fair evaluation. In the experiments, we conduct a comprehensive analysis on ACADEMICEVAL using several LLM baselines, and the results show that ACADEMICEVAL is a challenging long-context LLM benchmark. Insightful findings are also elucidated for potentially strengthening the long-context modeling capabilities of LLMs and inspiring future long-context LLM benchmarks to evaluate LLMs more flexibly and holistically.

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

# A   ADDITIONAL INFORMATION FOR EVALAUTION

## A.1   EVALUATION CRITERIA AND HYPERPARAMETERS

**API Access.**   In this paper, we conduct a comprehensive evaluation over ACADEMICEVAL benchmark using the LLM API provided by together.ai[6]. For each API call, we fix the temperature parameter to 0 (*i.e.*, greedy decoding).

**Input Truncation.**   By default, we use a BERT tokenizer to calculate the number of input tokens for ACADEMICEVAL. However, since the tokenizer of each LLM is usually different, it will cause some inputs to exceed the context length limit of the LLM. Therefore, for the evaluation of each LLM, we additionally download its tokenizer configuration file from the official website at hugging face, which is utilized to ensure correct and accurate truncation of input tokens.

**Refinement of LLM Responses.**   For the TITLE WRITING task, the responses of LLMs are relatively short. If the response contains some extra redundant information, it will have a greater impact on the evaluation metric score (although we have given LLM instructions not to generate irrelevant information). Therefore, for the TITLE WRITING task, we additionally refine the LLM responses, for example, removing irrelevant information such as "here is the title". For other tasks, since LLM's responses are relatively long, occasional small amounts of irrelevant information will not have a significant impact on the evaluation, so we do not perform any refinement on LLM's responses in this case.

**Details of the implementation of RALM.**   We use the inputs of ACADEMICEVAL as the external corpus of RALM (such as Target Content and Reference Content introduced in Section F). For text split, we use the RecursiveCharacterTextSplitter from LangChain[7] and set chunk size and chunk overlap to 512 and 64, respectively. For each retrieval, we recall up to 12 text chunks (limited by the context length of standard LLMs) based on text similarity (semantic similarity based on inner product for dense retrievers or similarity based on word frequency for sparse retrievers).

---

[6]https://www.together.ai/
[7]https://www.langchain.com/

## B  API COST

We adopt LLM API provided by together.ai[8] to conduct experiments in this paper. API costs mainly come from evaluating the test set of ACADEMICEVAL, which are estimated to be around $300.

---

[8]https://www.together.ai/

## C  LIMITATION AND FUTURE IMPROVEMENT

ACADEMICEVAL is a live benchmark without label leakage, which leverages co-author papers from a collected co-author graph as few-shot demonstrations to make the context length flexible and scalable. ACADEMICEVAL adopts arXiv as its data source without the need for manual labeling, and the content of the papers on it can naturally serve as high-quality and expert-curated annotations.

However, ACADEMICEVAL still has some limitations:

- **Task Diversity.** ACADEMICEVAL currently has only four academic writing tasks, which limits the task diversity.

- **Independent Evaluation of the Paper Section.** In ACADEMICEVAL, we independently evaluate the section content extracted from a paper, which may lack a comprehensive evaluation of the paper as a whole.

- **Popularity Bias.** ACADEMICEVAL first collects a co-author graph from arXiv, which contains a subset of all papers on arXiv. Therefore, the collected papers may have some popularity bias. For example, most of the papers may come from a few active authors, which will cause bias in the evaluation.

Based on the above limitations, our future improvements will include:

- **Introduce More Data Source.** The goal of ACADEMICEVAL is to make context length flexible and scalable by using few-shot demonstrations and high-quality labels without manual annotation, so papers on arXiv are a more suitable data source. We will consider adding other websites as data sources in the future, such as some question-answering websites (Stack Overflow or Reddit, etc.). In this case, we can use the best answers as high-quality labels. By modeling the citation relationship between posts into a graph, we can also obtain few-shot demonstrations to enrich the context length.

- **K-fold Cross-validation.** We can use k-fold cross-validation for a paper, that is, leaving a section (or fold) as the label, the remaining sections as inputs, and finally calculating the average of all leave-one-out evaluation scores.

- **Eliminate Popularity Bias.** We will perform probabilistic sampling on papers when collecting the co-author graph and give a lower sampling probability to active authors to alleviate the impact of popularity bias.

## D  SOCIAL IMPACT

The proposed benchmark ACADEMICEVAL will promote the academic community's exploration of using LLMs to automate academic writing tasks. Here are some key points highlighting its significance:

- **Efficiency and Productivity.** LLMs can drastically reduce the time and effort required for various academic writing tasks. These tasks include drafting papers, writing literature reviews, summarizing research articles, and generating bibliographies. By automating these processes, researchers can focus more on high-level thinking, experimentation, and analysis.

- **Enhanced Writing Quality.** LLMs have the ability to produce coherent and grammatically correct text, which can improve the overall quality of academic writing. They can assist in refining arguments, improving clarity, and ensuring consistency in style and tone, which is particularly useful for non-native English speakers.

- **Support for Multidisciplinary Research.** Given their training on diverse topics, LLMs can assist researchers in exploring interdisciplinary approaches by providing information and generating content across various fields of study. This can foster innovation and encourage collaboration between different academic disciplines.

# E  DETAILS OF LIVE LEADERBOARD

To enhance usability, we aim to create a live leaderboard on Hugging Face to help users easily utilize our benchmarks and compare various models. Our leaderboard will provide the following functionalities:

1. **Live updates and time selection:** We will update the dataset periodically to ensure it includes the latest papers from arXiv, which the LLMs have never seen before, to prevent label leakage. Users can choose the version they wish to use.

2. **Different abstraction tasks:** We will create separate leaderboards for each task. Users are welcome to run their models on one or more tasks and report their results.

3. **Ease of use:** We will provide detailed and standardized instructions so that users can easily run the pipeline with their models and obtain the results.

# F  LLM PROMPTS

In this section, we present the LLM prompts used in the experiments, including TITLE WRITING, ABSTRACT WRITING, INTRODUCTION WRITING, and RELATED WORK WRITING. For each academic writing task, we provide prompts for standard LLMs, long-context LLMs, and RALM (RALM additionally includes the retrieval query).

## F.1  LLM PROMPTS FOR TITLE WRITING

---

**Prompt for Standard and Long-context LLMs on TITLE-10K**

Please read the following Target Content carefully and summarize the Target Content as required.
### Target Content: {CONTENT}
### Target Content Abstract: {ABSTRACT}
Please craft a title highly summarizing the main theme from the above provided Target Content. The title should be of appropriate length (strictly limited to about 10 words). The title should also include and highlight the core and most critical theme of the Target Content, ignoring minor and redundant information. Please ensure that the title captures the essence of the Target Content in a clear and concise manner. Please output the title directly without including other redundant or irrelevant text.

---

**Prompt for Standard and Long-context LLMs on TITLE-30K and TITLE-31K-G**

Please read the following Reference Content and Output carefully and summarize the Target Content as required.
### Reference Content 0: {CONTENT_0}
### Reference Abstract 0: {ABSTRACT_0}
### Reference Output 0: {OUTPUT_0}

...
### Target Content: {CONTENT}
### Target Content Abstract: {ABSTRACT}
Please craft a title highly summarizing the main theme from the above provided Target Content. The Reference Content and Output provide some demonstrations, which may also contain some information that is potentially related to the Target Content. You can refer to the input and output text forms of the Reference Content and Output to assist in summarizing the Target Content and try to explore and use the information that is potentially related to the Target Content contained in the Reference Content and Output. The title should be of appropriate length (strictly limited to about 10 words). The title should also include and highlight the core and most critical theme of the Target Content, ignoring minor and redundant information. Please ensure that the title captures the essence of the Target Content in a clear and concise manner. Please output the title directly without including other redundant or irrelevant text.

---

**Prompt for RALM on TITLE-10K, TITLE-30K, and TITLE-31K-G**

Please read the following Target Content carefully and summarize the Target Content as required.
### Target Content 0: {CONTENT_0}
### Target Content 1: {CONTENT_1}
...
### Target Content Abstract: {ABSTRACT}
Please craft a title highly summarizing the main theme from all the above provided Target Contents. The title should be of appropriate length (strictly limited to about 10 words). The title should also include and highlight the core and most critical theme of the Target Contents, ignoring minor and redundant information. Please ensure that the title captures the essence of the Target Contents in a clear and concise manner. Please output the title directly without including other redundant or irrelevant text.

**Retrieval Query for RALM on TITLE-10K, TITLE-30K, and TITLE-31K-G**

Please craft a title highly summarizing the main theme of the provided text. The abstract of the text is: {ABSTRACT}

## F.2    LLM PROMPTS FOR ABSTRACT WRITING

**Prompt for Standard and Long-context LLMs on ABS-9K**

Please read the following Target Content carefully and summarize the Target Content as required.
### Target Content: {CONTENT}
### Target Content Title: {TITLE}
Please craft an abstract summarizing the key points from the above provided Target Content. The abstract should be of appropriate length (around 200 words) and include the main theme, significant findings or arguments, and conclusions of the Target Content. Please ensure that the abstract captures the essence of the Target Content in a clear, coherent, and succinct manner. Please output the abstract directly without including other redundant or irrelevant text.

**Prompt for Standard and Long-context LLMs on ABS-28K and ABS-29K-G**

Please read the following Reference Content and Output carefully and summarize the Target Content as required.
### Reference Content 0: {CONTENT_0}
### Reference Title 0: {TITLE_0}
### Reference Output 0: {OUTPUT_0}
...
### Target Content: {CONTENT}
### Target Content Title: {TITLE}
Please craft an abstract summarizing the key points from the above provided Target Content. The Reference Content and Output provide some demonstrations, which may also contain some information that is potentially related to the Target Content. You can refer to the input and output text forms of the Reference Content and Output to assist in summarizing the Target Content and try to explore and use the information that is potentially related to the Target Content contained in the Reference Content and Output. The abstract should be of appropriate length (around 200 words) and include the main theme, significant findings or arguments, and conclusions of the Target Content. Please ensure that the abstract captures the essence of the Target Content in a clear, coherent, and succinct manner. Please output the abstract directly without including other redundant or irrelevant text.

---

**Prompt for RALM on ABS-10K, ABS-30K, and ABS-31K-G**

Please read the following Target Content carefully and summarize the Target Content as required.
### Target Content 0: {CONTENT_0}
### Target Content 1: {CONTENT_1}
...
### Target Content Title: {TITLE}
Please craft an abstract summarizing the key points from all the above provided Target Contents. The abstract should be of appropriate length (around 200 words) and include the main theme, significant findings or arguments, and conclusions of the Target Contents. Please ensure that the abstract captures the essence of the Target Contents in a clear, coherent, and succinct manner. Please output the abstract directly without including other redundant or irrelevant text.

---

**Retrieval Query for RALM on ABS-10K, ABS-30K, and ABS-31K-G**

Please craft an abstract summarizing the key points of the provided text. The title of the text is: {TITLE}

---

### F.3 LLM PROMPTS FOR INTRODUCTION WRITING

---

**Prompt for Standard and Long-context LLMs on INTRO-8K**

Please read the following Target Content carefully and summarize the Target Content as required.
### Target Content: {CONTENT}
### Target Content Title: {TITLE}
### Target Content Abstract: {ABSTRACT}
Please craft an introduction summarizing the key points from the above provided Target Content. The introduction should be of appropriate length (about 1000 to 1500 words). The introduction should first describe the topic or main theme of the Target Content, then provide relevant background knowledge, and summarize the existing relevant research on this topic from the Target Content, point out their advantages and disadvantages, and highly summarize the specific research problem and problem statement targeted by the Target Content. Next, describe in detail the core approach or insights proposed by the Target Content on this topic and include any necessary experimental results. Then, use about 3 short paragraphs (each paragraph is about 50 words) to highly summarize the approach or insights proposed in the Target Content, as well as the experimental results. Finally, briefly give an overview of the Target Content's structure. Please ensure that the introduction captures the essence of the Target Content in a clear, coherent, and succinct manner. Please output the introduction directly without including other redundant or irrelevant text.

---

**Prompt for Standard and Long-context LLMs on INTRO-28K and INTRO-28K-G**

Please read the following Reference Content and Output carefully and summarize the Target Content as required.
### Reference Content 0: {CONTENT_0}
### Reference Title 0: {TITLE_0}
### Reference Abstract 0: {ABSTRACT_0}
### Reference Output 0: {OUTPUT_0}

...
### Target Content: {CONTENT}
### Target Content Title: {TITLE}
### Target Content Abstract: {ABSTRACT}
Please craft an introduction summarizing the key points from the above provided Target Content. The Reference Content and Output provide some demonstrations, which may also contain some information that is potentially related to the Target Content. You can refer to the input and output text forms of the Reference Content and Output to assist in summarizing the Target Content and try to explore and use the information that is potentially related to the Target Content contained in the Reference Content and Output. The introduction should be of appropriate length (about 1000 to 1500 words). The introduction should first describe the topic or main theme of the Target Content, then provide relevant background knowledge, and summarize the existing relevant research on this topic from the Target Content, point out their advantages and disadvantages, and highly summarize the specific research problem and problem statement targeted by the Target Content. Next, describe in detail the core approach or insights proposed by the Target Content on this topic and include any necessary experimental results. Then, use about 3 short paragraphs (each paragraph is about 50 words) to highly summarize the approach or insights proposed in the Target Content, as well as the experimental results. Finally, briefly give an overview of the Target Content's structure. Please ensure that the introduction captures the essence of the Target Content in a clear, coherent, and succinct manner. Please output the introduction directly without including other redundant or irrelevant text.

**Prompt for RALM on INTRO-8K, INTRO-28K, and INTRO-28K-G**

Please read the following Target Content carefully and summarize the Target Content as required.
### Target Content 0: {CONTENT_0}
### Target Content 1: {CONTENT_1}

...
### Target Content Title: {TITLE}
### Target Content Abstract: {ABSTRACT}
Please craft an introduction summarizing the key points from all the above provided Target Contents. The introduction should be of appropriate length (about 1000 to 1500 words). The introduction should first describe the topic or main theme of the Target Contents, then provide relevant background knowledge, summarize the existing relevant research on this topic from the Target Contents, point out their advantages and disadvantages, and highly summarize the specific research problem and problem statement targeted by the Target Contents. Next, describe in detail the core approach or insights proposed by the Target Contents on this topic and include any necessary experimental results. Then, use about 3 short paragraphs (each paragraph is about 50 words) to highly summarize the approach or insights proposed in the Target Contents, as well as the experimental results. Finally, briefly give an overview of the Target Contents' structure. Please ensure that the introduction captures the essence of the Target Contents in a clear, coherent, and succinct manner. Please output the introduction directly without including other redundant or irrelevant text.

---

**Retrieval Query for RALM on INTRO-8K, INTRO-28K, and INTRO-28K-G**

Please craft an introduction summarizing the main theme of the provided text (including background knowledge, advantages and disadvantages of existing research and challenges, the proposed approach, experimental results, etc.). The title of the text is {TITLE}. The abstract of the text is {ABSTRACT}.

---

### F.4 LLM PROMPTS FOR RELATED WORK WRITING

---

**Prompt for Standard and Long-context LLMs on RELATED-34K**

Please read the following Target Content and Target Citations carefully and summarize the Target Citations according to the topic of the Target Content as required.
### Target Citation 0:
Target Citation Title: {C_TITLE_0}
Target Citation Abstract: {C_ABSTRACT_0}

...
### Target Content: {CONTENT}
### Target Content Title: {TITLE}
### Target Content Abstract: {ABSTRACT}
Given the Target Content and its Abstract and Title, along with its Target Citations (including Target Citation Title and Abstract), please craft a related work summarizing the key points from the above provided Target Citations. There is no specific length requirement or limit for the entire related work (it is best to keep it around 500 to 1000 words), but each Target Citation that appears in the related work needs to be highly summarized in extremely concise and short sentences. You can refer to the topic or main theme described by the Target Content and its Abstract and Title to filter irrelevant information in the Target Citations and leverage relevant information. Furthermore, you can categorize the relevant Target Citations, briefly summarize the advantages and disadvantages of each categorization, and explain the advantages of the approach proposed in the Target Content. Please ensure that the related work captures all the relevant key points of the Target Citations in a clear, coherent, and succinct manner. Please output the related work directly without including other redundant or irrelevant text.

---

---

**Prompt for Standard and Long-context LLMs on RELATED-53K and RELATED-53K-G**

Please read the following Reference Content and Output carefully and summarize the Target Citations according to the topic of the Target Content as required.
### Reference Content 0: {CONTENT_0}
### Reference Title 0: {TITLE_0}
### Reference Abstract 0: {ABSTRACT_0}
### Reference Output 0: {OUTPUT_0}

...
### Target Citation 0:
Target Citation Title: {C_TITLE_0}
Target Citation Abstract: {C_ABSTRACT_0}

...
### Target Content: {CONTENT}
### Target Content Title: {TITLE}
### Target Content Abstract: {ABSTRACT}
Given the Target Content and its Abstract and Title, along with its Target Citations (including Target Citation Title and Abstract), please craft a related work summarizing the key points from the above provided Target Citations. The Reference Content and Output provide some demonstrations, which may also contain some information that is potentially related to the Target Content. You can refer to the input and output text forms of the Reference Content and Output to assist in summarizing the Target Citations and try to explore and use the information (e.g., related citations missing from the Target Citations) that is potentially related to the Target Content contained in the Reference Content and Output. There is no specific length requirement or limit for the entire related work (it is best to keep it around 500 to 1000 words), but each Target Citation that appears in the related work needs to be highly summarized in extremely concise and short sentences. You can refer to the topic or main theme described by the Target Content and its Abstract and Title to filter irrelevant information in the Target Citations and leverage relevant information. Furthermore, you can categorize the relevant Target Citations, briefly summarize the advantages and disadvantages of each categorization, and explain the advantages of the approach proposed in the Target Content. Please ensure that the related work captures all the relevant key points of the Target Citations in a clear, coherent, and succinct manner. Please output the related work directly without including other redundant or irrelevant text.

---

**Prompt for RALM on RELATED-34K, RELATED-53K, and RELATED-53K-G**

Please read the following Target Content and Target Citations carefully and summarize the Target Citations according to the topic of the Target Content as required.
### Target Content 0: {CONTENT_0}
### Target Content 1: {CONTENT_1}

...
### Target Content Title: {TITLE}
### Target Content Abstract: {ABSTRACT}
Given the Target Content Abstract and Title, please craft a related work summarizing the key points from all the above provided Target Contents. There is no specific length requirement or limit for the entire related work (it is best to keep it around 500 to 1000 words), but each Target Content that appears in the related work needs to be highly summarized in extremely concise and short sentences. You can refer to the topic or main theme described by the Target Content Abstract and Title to filter irrelevant information in the Target Contents and leverage relevant information. Furthermore, you can categorize the relevant Target Contents and briefly summarize the advantages and disadvantages of each categorization. Please ensure that the related work captures all the relevant key points of the Target Contents in a clear, coherent, and succinct manner. Please output the related work directly without including other redundant or irrelevant text.

> **Retrieval Query for RALM on RELATED-34K, RELATED-53K, and RELATED-53K-G**
>
> Please craft a related work summarizing all the relevant key points of the provided text. The title of the text is {TITLE}. The abstract of the text is {ABSTRACT}.

## G EXAMPLES

In this section, we present some generation examples w.r.t. TITLE WRITING, ABSTRACT WRITING, and INTRODUCTION WRITING tasks.

### G.1 TITLE WRITING

---

**Comparison between Generated Title and Ground Truth on TITLE WRITING Task**

**Ground Truth:**
When to Trust LLMs: Aligning Confidence with Response Quality

• **Qwen 1.5 Chat (72B)**

**Generated Title (0-shot):**
CONQORD: Reinforcement Learning for Confidence-Response Alignment in Large Language Models

**Generated Title (2-shot demonstrations from random selection):**
CONQORD: Enhancing LLM Confidence Alignment for Improved Trustworthiness and Retrieval

**Generated Title (2-shot demonstrations from co-author papers):**
CONQORD: Enhancing Confidence Calibration for Language Models via Reinforcement Learning

• **Mixtral-8x7B Instruct (46.7B)**

**Generated Title (0-shot):**
CONQORD: Aligning Confidence with Quality in Large Language Models

**Generated Title (2-shot demonstrations from random selection):**
CONQORD: Reinforcement Learning for Confidence-Quality Alignment in LLMs

**Generated Title (2-shot demonstrations from co-author papers):**
CONQORD: Aligning Confidence with Response Quality in LLMs

• **Nous Hermes 2 - Mixtral 8x7B-DPO (46.7B)**

**Generated Title (0-shot):**
Aligning Confidence with Response Quality in Large Language Models using Reinforcement Learning with Dual-Component Reward Function: CONQORD Approach

**Generated Title (2-shot demonstrations from random selection):**
Aligning Confidence with Response Quality in Large Language Models: Introducing CONQORD

**Generated Title (2-shot demonstrations from co-author papers):**
Confidence Alignment in Large Language Models via Reinforcement Learning: A CONQORD Approach for Trust

---

**Comparison between Generated Title and Ground Truth on TITLE WRITING Task**

**Ground Truth:**
Distillation for Multilingual Information Retrieval

● **Qwen 1.5 Chat (72B)**

**Generated Title (0-shot):**
Multilingual Translate-Distill: Enhancing Cross-Language Information Retrieval Models

**Generated Title (2-shot demonstrations from random selection):**
Multilingual Translate-Distill for Enhanced Cross-Language Information Retrieval

**Generated Title (2-shot demonstrations from co-author papers):**
Multilingual Translate-Distill for Effective Multilingual Information Retrieval

● **Mixtral-8x7B Instruct (46.7B)**

**Generated Title (0-shot):**
Multilingual Translate-Distill for Cross-Language Information Retrieval

**Generated Title (2-shot demonstrations from random selection):**
Multilingual Translate-Distill for Cross-Language Information Retrieval

**Generated Title (2-shot demonstrations from co-author papers):**
Multilingual Translate-Distill for MLIR: Extending Translate-Distill for Robust Language Mixing

● **Nous Hermes 2 - Mixtral 8x7B-DPO (46.7B)**

**Generated Title (0-shot):**
Multilingual Translate-Distill: Enhancing Multilingual Information Retrieval with Translation and Distillation

**Generated Title (2-shot demonstrations from random selection):**
Multilingual Translate-Distill: Efficient Training for Multilingual Information Retrieval

**Generated Title (2-shot demonstrations from co-author papers):**
Multilingual Translate-Distill: Training Dual-Encoder Models for Multilingual Information Retrieval

## G.2 ABSTRACT WRITING

---

**Comparison between Generated Abstract and Ground Truth on ABSTRACT WRITING Task**

**Ground Truth:**
In recent years, neural ranking models (NRMs) have been shown to substantially out-perform their lexical counterparts in text retrieval. In traditional search pipelines, a combination of features leads to well-defined behavior. However, as neural approaches become increasingly prevalent as the final scoring component of engines or as standalone systems, their robustness to malicious text and, more generally, semantic perturbation needs to be better understood. We posit that the transformer attention mechanism can induce exploitable defects through positional bias in search models, leading to an attack that could generalize beyond a single query or topic. We demonstrate such defects by showing that non-relevant text–such as promotional content–can be easily injected into a document without adversely affecting its position in search results. Unlike previous gradient-based attacks, we demonstrate these biases in a query-agnostic fashion. In doing so, without the knowledge of topicality, we can still reduce the negative effects of non-relevant content injection by controlling injection position. Our experiments are conducted with simulated on-topic promotional text automatically generated by prompting LLMs with topical context from target documents. We find that contextualization of a non-relevant text further reduces negative effects whilst likely circumventing existing content filtering mechanisms. In contrast, lexical models are found to be more resilient to such content injection attacks. We then investigate a simple yet effective compensation for the weaknesses of the NRMs in search, validating our hypotheses regarding transformer bias.

● **Qwen 1.5 Chat (72B)**

**Generated Abstract (2-shot demonstrations from co-author papers):**
The study investigates the vulnerability of Neural Ranking Models (NRMs) to adversarial attacks through the injection of promotional content into documents. The authors propose a framework for generating contextualised text using Large Language Models (LLMs) and examine the impact of position and context on the relevance estimation of injected text. They find that NRMs, particularly those based on BERT and T5 architectures, are sensitive to the position of injected text, with content placed after salient sentences having a lesser negative effect on relevance. The study also introduces a mitigation strategy using a classifier to detect promotional content, demonstrating its effectiveness in reducing the negative impact on retrieval performance. The findings highlight the potential for exploiting positional bias in NRMs and the need for robustness against such attacks in semantic search.

● **Nous Hermes 2 - Mixtral 8x7B-DPO (46.7B)**

**Generated Abstract (2-shot demonstrations from co-author papers):**
This study investigates the exploitation of positional bias in neural ranking models (NRMs) for query-agnostic generative content injection in search. The authors propose the concept of attention bleed-through, where positive attention scores from relevant spans propagate to non-relevant spans, affecting relevance estimation. They find that injecting non-relevant text close to document sentences with high similarity scores with the query can largely affect the rank of documents. The study also demonstrates that controlling the position and contextualization of promotional text can mitigate its negative impact on retrieval performance. A model-agnostic mitigation strategy is proposed, which improves nDCG@10 significantly under a classic evaluation setting by reducing the effect of contextualization. The findings highlight the importance of addressing positional bias in NRMs and the potential risks of generative content injection in semantic search.

---

**Comparison between Generated Abstract and Ground Truth on ABSTRACT WRITING Task**

**Ground Truth:**
Developing autonomous agents for mobile devices can significantly enhance user inter-actions by offering increased efficiency and accessibility. However, despite the growing interest in mobile device control agents, the absence of a commonly adopted benchmark makes it challenging to quantify scientific progress in this area. In this work, we introduce B-MoCA, a novel benchmark designed specifically for evaluating mobile device control agents. To create a realistic benchmark, we develop B-MoCA based on the Android operating system and define 60 common daily tasks. Importantly, we incorporate a ran-domization feature that changes various aspects of mobile devices, including user interface layouts and language settings, to assess generalization performance. We benchmark diverse agents, including agents employing large language models (LLMs) or multi-modalLLMs as well as agents trained from scratch using human expert demonstrations. While these agents demonstrate proficiency in executing straightforward tasks, their poor performance on complex tasks highlights significant opportunities for future research to enhance their effectiveness. Our source code is publicly available at https://b-moca.github.io.

• **Qwen 1.5 Chat (72B)**

**Generated Abstract (2-shot demonstrations from co-author papers):**
The Benchmark for Mobile Device Control Agents (B-MoCA) is a novel evaluation platform designed to assess the performance of agents in controlling mobile devices across diverse configurations. The benchmark features 60 practical tasks grounded in real-life scenarios and supports randomization of device attributes like icon placements, wallpapers, languages, and device types to test generalization ability. The study compares three types of agents: LLM (Large Language Model) agents, MLLM (Multimodal Large Language Model) agents, and Vision-Language-UI (VLUI) agents. Results show that while LLM and MLLM agents perform well in simple tasks, they struggle with complex tasks requiring sequential decision-making. VLUI agents, trained from scratch, demonstrate better performance in completing tasks but still face challenges with severe visual changes in unseen device configurations. The study highlights the importance of diverse training data and the need for more efficient methods to leverage multi-modal inputs. Overall, B-MoCA provides a comprehensive framework for evaluating and advancing mobile device control agents in real-world settings.

• **Nous Hermes 2 - Mixtral 8x7B-DPO (46.7B)**

**Generated Abstract (2-shot demonstrations from co-author papers):**
This paper introduces B-MoCA, a benchmark designed to evaluate the performance of mobile device control agents on diverse device configurations in executing common daily tasks. The benchmark includes 60 basic tasks grounded in realistic scenarios and a randomization feature that changes icon placements and sizes, wallpapers, languages, and device types. Baseline agents include LLM agents, MLLM agents, and Vision-Language-UI (VLUI) agents. Experiments reveal that LLM agents and MLLM agents show high robustness across diverse device configurations while they fall short on multiple sequential decision-making. Agents trained with BC successfully mimic expert behaviors but lack generalization ability in test environments with unseen device configurations. The study highlights the limitations of existing methods in mobile device control and calls for future research.

### G.3 INTRODUCTION WRITING

---

**Comparison between Generated Introduction and Ground Truth on INTRODUCTION WRITING Task**

**Ground Truth:**
The original text is too long to be included here. Please see 2404.17196v1 via arXiv for more details.

- **Qwen 1.5 Chat (72B)**

**Generated Introduction (2-shot demonstrations from co-author papers):**
Retrieval poisoning is a newly identified threat to large language model (LLM)-powered applications, particularly those utilizing the retrieval augmented generation (RAG) technique. This threat arises from the vulnerability in LLM application frameworks, which allow attackers to subtly manipulate the content of documents in a way that appears benign to humans but can guide the LLMs to generate incorrect or malicious responses. The attack is executed by injecting invisible attack sequences into the documents, which remain undetected during the RAG process, leading to compromised application responses. The prevalence of LLM application frameworks, such as LangChain, has enabled the development of numerous LLM-powered applications. However, these frameworks often overlook the security risks associated with external content. Retrieval poisoning exploits this vulnerability by crafting malicious documents that, despite containing accurate information, can prompt the LLMs to produce misleading responses when used as reference sources during RAG. The success rate of such attacks is alarming, with preliminary experiments demonstrating an 88.33% success rate in misleading LLMs and a 66.67% success rate in real-world applications. To execute retrieval poisoning, attackers analyze the LLM application framework, identifying exploitable components like document parsers, text splitters, and prompt templates. They then generate an attack sequence that is invisible to human perception but influences the LLM's response generation. The crafted malicious document is designed to maintain its original appearance while containing the hidden attack sequence, which is strategically placed to remain undivided by text splitters. The proposed approach involves a two-step process: analyzing the framework to identify invisible injection points and generating the attack sequence. Attackers leverage the framework's prompt templates to create augmented requests that guide the LLMs towards generating targeted malicious responses. The effectiveness of the attack is demonstrated through experiments on different LLMs, augmented requests, and real-world applications, showcasing the potential impact of retrieval poisoning. In summary, retrieval poisoning is a stealthy attack method that exploits the vulnerabilities in LLM application frameworks, particularly in the RAG process. By injecting invisible attack sequences into documents, attackers can manipulate LLMs into generating incorrect responses, posing a significant security concern for LLM-powered applications. The proposed approach highlights the need for better understanding of LLM application frameworks and the development of more robust defense mechanisms to mitigate such threats. The paper structure delves into the problem, presents the attack methodology, demonstrates its effectiveness through experiments, and discusses potential defense strategies, ultimately contributing to the ongoing research on securing LLM-powered applications.

**Comparison between Generated Introduction and Ground Truth on INTRODUCTION WRITING Task**

**Ground Truth:**
The original text is too long to be included here. Please see 2404.18465v2 via arXiv for more details.

● **Qwen 1.5 Chat (72B)**

**Generated Introduction (2-shot demonstrations from co-author papers):**
The M3oE framework is a pioneering solution to the challenge of multi-domain multi-task recommendation, addressing the intricate dependencies among multiple domains and tasks in a disentangled and adaptive manner. Recognizing the need for a more comprehensive approach in practical recommendation systems, M3oE integrates multi-domain information and leverages knowledge transfer across domains and tasks, optimizing for multiple objectives simultaneously. The framework consists of three mixture-of-experts modules, each dedicated to capturing common, domain-specific, and task-specific user preferences. A two-level fusion mechanism is employed to effectively extract and integrate features across diverse domains and tasks, ensuring a balanced representation of user preferences. Existing multi-domain and multi-task recommendation methods have shown promise in enhancing recommendation accuracy by exploiting shared patterns. However, they often fall short when faced with the complexity of simultaneously addressing multiple domains and tasks. M3oE stands out by introducing a self-adaptive approach, utilizing AutoML to optimize the model structure dynamically. This adaptability is crucial for real-world scenarios where data characteristics and task requirements can vary significantly. The core of M3oE lies in its three expert modules: shared, domain, and task experts. The shared expert module learns common patterns across domains and tasks, while the domain expert module focuses on domain-specific characteristics. The task expert module, on the other hand, models task-specific user preferences. The two-level fusion mechanism balances the contribution of these modules, allowing for a flexible and effective representation of user preferences. Experimental results on benchmark datasets demonstrate M3oE's superior performance compared to state-of-the-art baselines, highlighting its ability to address the multi-domain multi-task seesaw problem. In summary, M3oE is a novel and adaptive multi-domain multi-task recommendation framework that integrates multi-domain information, transfers knowledge across domains and tasks, and optimizes for multiple objectives. The framework's disentangled and fused expert modules, along with the AutoML-driven fusion weights, enable it to effectively capture user preferences in complex recommendation scenarios. The proposed approach not only surpasses existing methods in performance but also offers a more generalizable solution to the multi-domain multi-task recommendation challenge. The structure of the paper delves into the problem definition, framework overview, detailed module descriptions, experimental setup, and results, providing a comprehensive understanding of M3oE's design and effectiveness.

