# OpenReview forum: "AcademicEval: Live Long-Context LLM Benchmark"
_ICLR.cc/2025/Conference — ICLR 2025 Conference Withdrawn Submission_

### Official Review · Reviewer_MQBQ · 2024-11-04

**Soundness:** 2
**Presentation:** 3
**Contribution:** 2
**Rating:** 3
**Confidence:** 3

**Summary:**

The authors present a benchmark to evaluate models on their ability to perform long context generation tasks. They used ArXiv as their main data source and they setup 4 different tasks to evaluate on, they are TITLE, ABSTRACT, INTRODUCTION, and RELATED WORK. The LLM would take a co-author graph that is generated via the arXiv API and draw on the co-author papers as informative few-shot demonstrations. The generations are evaluate based on BERTScore, and RougeL.

**Strengths:**

- The motivation of testing LLMs ability to perform long context text-generation tasks is great
- Writing different sections of a paper is a nice challenge for this kind of task.
- Having a benchmark that requires no human labor is beneficial
- The idea of using live evaluation to avoid label leakage is decent

**Weaknesses:**

- While the benchmark is about long context text-generation, focusing on generating the 4 subsections of Arxiv makes the benchmark not comprehensive enough for evaluating how well an LLM performs in  the grandiose task of long context text-generation. There are so many other long context text-generation tasks that require different type of reasoning and setup. The title might need to be qualified to something like "evaluating llms on a subset of research paper generation". How do you ensure this benchmark generalize to other long-context generation task?

- RougeL and BERTScore don't seem like they are good measures for evaluating whether the generated text is good enough. BERT in BERTScore has been trained on extracting embeddings from small chunks of text to identify things like sentiment, and RougeL checks for word-for-word overlap which doesn't make sense in these ArXiv generation tasks since there are so many different ways we can semantically write the same thing for the introduction. Why would BERTScore be a good measure to test whether two "Introductions" are similar? One valid possible metric is G-Eval which was used to evaluate whether two free-form texts are similar, which seems to be valid in this case. Have you considered such a metric?

- There needs to be human evaluation to see if BERTScore matches with how humans would judge a paper, I would say that is the only way to know if the BERTScore measure makes sense for these tasks. Have you considered such a process?

- For the related work section, there are so many permutations and subsets of the literature that can be cited and still would make the section valid. Also, there are so many ways to relate the paper with the paper at hand - how do you make sure those type of generations are not penalized just because they are not almost word for word of the groundtruth data?

- While the idea of using live evaluation is nice, evaluating whether a new model is good in comparison to the older ones would require us to re-evaluate all the models on the new data which makes things costly and unscalable. A better approach would be to have a hidden test set that is not in ArXiv to make sure that the models would never have never seen them (avoiding label leakage). That way when we have a new model, we can just evaluate it and add its score to the benchmark. Fixed benchmarks are always better in this regard of having a scalable, consistent test bed.

**Questions:**

What are your thoughts on the weaknesses above?

---

### Official Review · Reviewer_zFmj · 2024-11-04

**Soundness:** 3
**Presentation:** 3
**Contribution:** 2
**Rating:** 3
**Confidence:** 2

**Summary:**

The paper proposes ACADEMICEVAL, a benchmark aimed at evaluating the capabilities of Large Language Models (LLMs) in handling long-context academic writing tasks. Utilizing academic papers from arXiv as its primary data source, the benchmark is designed to assess LLM performance across different levels of abstraction in tasks such as title generation, abstract writing, and related work summarization. ACADEMICEVAL emphasizes flexibility in context length and features live evaluation, which aims to mitigate label leakage issues.

**Strengths:**

1 - By using arXiv papers, ACADEMICEVAL leverages readily available, high-quality academic content, which reduces reliance on labor-intensive manual annotation.

2 - The benchmark’s periodic updates from arXiv mitigate risks associated with label leakage and maintain relevance in LLM assessment.

3 - The benchmark includes multiple academic writing tasks with different abstraction levels, offering a broad evaluation framework for long-context generation.

**Weaknesses:**

1 - The proposed benchmark does not introduce any fundamentally novel insights or methodological contributions for evaluating long-context LLMs, instead reusing existing concepts (e.g., hierarchical task structure and few-shot learning demonstrations).

2 - ACADEMICEVAL focuses on a narrow set of tasks related to academic writing, which limits its applicability and fails to test LLMs across a wider range of real-world long-context scenarios.

3 - The paper does not adequately demonstrate the limitations of existing benchmarks to justify the need for ACADEMICEVAL. While the authors critique current benchmarks for rigid context limits and label leakage, these issues are only partially addressed by ACADEMICEVAL’s live evaluation and flexible context lengths.

**Questions:**

How does ACADEMICEVAL’s hierarchical structure provide insights that are not achievable with existing benchmarks like LongBench or L-Eval?

Can the live evaluation feature sufficiently prevent model contamination, given the rapid development of LLMs and their potential exposure to recent datasets?

Have any user studies been conducted to validate the effectiveness of few-shot demonstrations from the co-author graph in improving LLM performance on long-context tasks?

---

### Official Review · Reviewer_gEox · 2024-11-04

[review text omitted: it was posted to a different submission]

---

> ### Author Response · Authors · 2024-11-14
> **Wrong Review**
>
> Thanks for your feedback, but it seems to be the reviews from other submissions.

---

### Note · Authors · 2024-12-15

I have read and agree with the venue's withdrawal policy on behalf of myself and my co-authors.